# Deuterated Glutamate-Mediated Neuronal Activity on Micro-Electrode Arrays

**DOI:** 10.3390/mi11090830

**Published:** 2020-08-31

**Authors:** Wataru Minoshima, Kyoko Masui, Tomomi Tani, Yasunori Nawa, Satoshi Fujita, Hidekazu Ishitobi, Chie Hosokawa, Yasushi Inouye

**Affiliations:** 1AIST–Osaka University Advanced Photonics and Biosensing Open Innovation Laboratory, National Institute of Advanced Industrial Science and Technology, AIST, Osaka 565-0871, Japan; w.minoshima@aist.go.jp (W.M.); masui-k@aist.go.jp (K.M.); nawa.y@aist.go.jp (Y.N.); s-fujita@aist.go.jp (S.F.); ishitobi@ap.eng.osaka-u.ac.jp (H.I.); 2Graduate School of Frontier Biosciences, Osaka University, Osaka 565-0871, Japan; 3Biomedical Research Institute, National Institute of Advanced Industrial Science and Technology (AIST), Ikeda 563-0026, Japan; tomomi-tani@aist.go.jp; 4Department of Applied Physics, Graduate School of Engineering, Osaka University, Osaka 565-0871, Japan; 5Department of Chemistry, Division of Molecular Materials Science, Graduate School of Science, Osaka City University, Osaka 558-8585, Japan

**Keywords:** cultured neuronal network, spontaneous activity, microelectrode array, neurotransmitter, glutamate receptor, deuterium-labeled glutamate

## Abstract

The excitatory synaptic transmission is mediated by glutamate (GLU) in neuronal networks of the mammalian brain. In addition to the synaptic GLU, extra-synaptic GLU is known to modulate the neuronal activity. In neuronal networks, GLU uptake is an important role of neurons and glial cells for lowering the concentration of extracellular GLU and to avoid the excitotoxicity. Monitoring the spatial distribution of intracellular GLU is important to study the uptake of GLU, but the approach has been hampered by the absence of appropriate GLU analogs that report the localization of GLU. Deuterium-labeled glutamate (GLU-D) is a promising tracer for monitoring the intracellular concentration of glutamate, but physiological properties of GLU-D have not been studied. Here we study the effects of extracellular GLU-D for the neuronal activity by using primary cultured rat hippocampal neurons that form neuronal networks on microelectrode array. The frequency of firing in the spontaneous activity of neurons increased with the increasing concentration of extracellular GLU-D. The frequency of synchronized burst activity in neurons increased similarly as we observed in the spontaneous activity. These changes of the neuronal activity with extracellular GLU-D were suppressed by antagonists of glutamate receptors. These results suggest that GLU-D can be used as an analog of GLU with equivalent effects for facilitating the neuronal activity. We anticipate GLU-D developing as a promising analog of GLU for studying the dynamics of glutamate during neuronal activity.

## 1. Introduction

The neuronal network in dissociated culture of neurons has been used as a model system for studying the neuronal network in the brain [1,2,3,4], as the system shows the mechanisms that are essential for learning, memory, and cognition [5,6,7,8]. Spontaneous firing of action potentials in neurons can be observed in cultured networks [9,10,11,12,13], and that has been reported to be one of the bases for the brain functions [14,15,16].

In order to examine the spontaneous activity of neurons in neuronal networks, a microelectrode array (MEA) has been used, which is composed of a grid-shape arrangement of multiple planar microelectrodes fabricated on a substrate. The MEA device has been widely used for recording the spontaneous activity and evoked responses driven by electrical stimuli of primary neuronal cell cultures and has revealed the developmental changes of neuronal networks [17], stimulus-induced plasticity [18,19], and the various of pharmacological tests for neuronal activities [20,21,22]. The spikes produced by neuronal populations can be simultaneously measured by electrodes as extracellular field potentials. By using MEA, previous studies have shown that the spontaneous firing rates of cultured neurons reflected the neuronal activity that was altered by the concentration change of extracellular glucose [23]. Moreover, array-wide synchrony of spontaneous neuronal activity can be analyzed with MEA by simultaneously monitoring the extracellular potentials of synaptically connected multiple neurons [17,18,19]. MEA is a useful tool to monitor the activity of neuronal networks that is altered by the conditions of extracellular environment.

In the hippocampus, the balance of activities in excitatory and inhibitory synapses is strictly regulated for the control of normal neuronal activity [24]. An excitatory synaptic transmission is mediated by glutamate (GLU). Large numbers of glutamate receptors are expressed in synapses, and the spontaneous activity of neurons in vivo is known to be enhanced by extracellular GLU that binds to the receptors [25]. The frequency of spontaneous activity increased with increasing extracellular concentrations of GLU in dissociated neuronal networks cultivated on MEA [26]. As described above, GLU has important roles to maintain neuronal activity, but long-term exposure to GLU induces excitotoxicity to neurons.

In neuronal networks, NMDA-type glutamate receptors activate the neuronal nitric oxide synthase (nNOS), producing toxic levels of nitric oxide (NO). NMDA receptors and nNOS are associated with the post-synaptic density proteins. Overactivation of NMDA receptors under excitotoxic conditions mediates the Ca^2+^ influx, and that results in activation of nNOS via calmodulin [27,28]. The NO production leads to mitochondoria disfunction and superoxide production, and NO-superoxide interactions cause apotosis for neuronal cells [29]. Also, the nNOS is primarily expressed in the hippocampus and cortex [30]. To avoid excitotoxicity, excess GLU is retrieved by neuronal and glial cells to avoid the cell death in the nervous system [31,32], and some part of retrieved GLU is recycled to the synaptic vesicles of presynaptic neurons. So, the uptake of GLU by neuronal/glial cells is considered to be an important event that reflects the activity of neuronal network. Previous studies have reported that exocytosed GLU could be measured by using enzyme-amperometric detection [33,34,35] and fluorescence imaging of probes that specifically bind to GLU [36,37]. In spite of the usability of these methods, the previous methods cannot be applied to measure the concentration of intracellular GLU.

In order to visualize the localization of GLU both in extra- and intra-cellular space, GLU needs to be detected directly. Isotope-labeled chemicals have been utilized as stable tracers for studying the metabolism of biological molecules in the cells. For example, deuterium-labeled compounds have been used to trace the intermediate products during the metabolism [38]. Substitution of one or more forms of carbon-bonded hydrogen with deuterium gives isotope effects, and that imposes negligible steric effects and minimum influences on the physiological and chemical properties [39]. Isotope effects of deuterium have been widely used for tracing cellular metabolisms measured with X-ray irradiation [40], nuclear magnetic resonance [41,42], and NADPH analysis [43]. Also, a deuterium-labeled tracer could be used in optical measurements such as Raman spectroscopy [44,45,46]. In neuronal networks, deuterium-labeled glutamate (GLU-D) is expected to work as an analog of GLU for intracellular tracing, but the physiological studies are missing for testing if GLU-D is functionally equivalent to GLU in neuronal activity.

In the present study, we evaluated the effects of extracellular GLU-D on spontaneous neuronal activity of primary cultured rat hippocampal neurons. The spontaneous activity of neurons was measured with MEA in culture media with different extracellular concentrations of GLU and GLU-D. We used specific blockers of glutamate receptors for testing the effects of GLU-D on the spontaneous neuronal activity of neurons cultured on MEA.

## 2. Materials and Methods

### 2.1. Primary Cultured Hippocamal Neuronal Network on MEA

All animal experiments were conducted with the approval of the Committee for the Experiments involving Animals of National Institute of Advanced Industrial Science and Technology (AIST). Pregnant female Wistar rat (Slc:Wistar, Japan SLC, Shizuoka, Japan) was anesthetized with isoflurane (Pfizer, New York, NY, USA) and immediately sacrificed by cervical dislocation. Next, 18 days-old embryos were extracted and sacrificed under ice-cold Dulbecco’s phosphate buffered saline (D-PBS(-), FUJIFILM Wako Pure Chemical, Osaka, Japan). The hippocampi isolated from the embryos were cut into small pieces and treated with 0.25 *w*/*v*% trypsin-EDTA (Thermo Fisher Scientific, Waltham, MA, USA) solution at 37 °C for 15 min for digestion. The hippocampal cells were dissociated into single cells by mechanical pipetting. Dissociated cells were plated at the initial density of 2.5 × 10^5^ cells/cm^2^ on the MEA dish (MED-probe, MED-P515A, alpha MED scientific, Osaka, Japan, Figure 1a) coated with 0.02 *w*/*v*% polyethyleneimine (Sigma-Aldrich, St. Louis, MO, USA). On the MED probe, 50 μm × 50 μm of planar 64 microelectrodes were fabricated with 8 × 8 grid patterns at 150 μm-distances (center to center) on a glass substrate. In order to place the hippocampal cells on the area of microelectrodes, we used a cloning ring with 7 mm-inner diameter (Iwaki, Tokyo, Japan). The Dulbecco’s Modified Eagle Medium (Thermo Fisher Scientific) premixed cocktail of 5 μg/mL Insulin (Sigma-Aldrich, St. Louis, MO, USA), 100 units-100 μg/mL Penicillin-Streptomycin (Thermo Fisher Scientific), 5% heat-inactivated Fetal Bovine Serum (Thermo Fisher Scientific), and 5% heat-inactivated Horse Serum (Thermo Fisher Scientific) was used as a DMEM medium. Cells were cultured in a 37 °C incubator with 5% CO_2_/95% air for 22 to 37 days in vitro (DIV) in the mixed medium with the equal amount of fresh DMEM medium and the used DMEM medium conditioned for three days with glial cells. A half volume of the mixed medium was replaced with a new one every 2–3 days. Densely cultured neurons on the MED probe are shown in Figure 1b.

### 2.2. Control of GLU and GLU-D Concentrations and Pharamacological Treatments

Figure 2 shows the chemical structure of deuterium-exchanged L-glutamate (GLU-D, L-glutamate-d5, Cambridge Isotope Laboratories, Inc., Tewksbury, MA, USA). GLU (L-Glutamate, stock concentration 50 mM, Sigma-Aldrich) and GLU-D (stock concentration, 50 mM) in PBS were prepared. Afterwards, 1 mL of stock solution was introduced onto a quartz substrate of 12 mm-diameter attached to the bottom surface of a plastic culture dish (SF-S-D12, Fine Plus International Ltd., Kyoto, Japan) for Raman spectroscopy.

We prepared the recording medium that was consisted of 130 NaCl, 3 KCl, 2 CaCl_2_, 1 MgCl_2_, 10 HEPES, and 10 Glucose (in mM), and pH was adjusted to 7.3. The culture medium was replaced with the recording medium containing GLU or GLU-D. Concentrations of GLU and GLU-D in the stock media were adjusted to 10 mM in the recording medium. By adding 2, 4, and 6 μL of the stock solutions to 2 mL of recording medium, we adjusted the final concentration of 10, 20, and 30 μM of GLU/GLU-D in a cultured neuron to evaluate the changes of spontaneous activities from initial number at each GLU/GLU-D concentration. After the replacement of extracellular solutions, the primary cultures of neurons were incubated for 10 min in the recording medium and then the spontaneous activity of neurons was recorded.

For pharmacological experiments, we measured spontaneous activity at 10, 20, and 30 µM of GLU/GLU-D with glutamate receptors in a cultured neuron. We used AMPA-type glutamate receptors antagonist 6-cyano-7-nitroquinoxaline-2,3-dione (CNQX, Sigma-Aldrich) and NMDA-type glutamate receptors antagonist (2R)-amino-5-phosphonovaleric acid (D-APV, Sigma-Aldrich). 10 mM of CNQX and 50 mM of D-APV in the recording medium were prepared for stock solutions. 2 µL of each stock solution was added to 2 mL of recording medium in MED probe to make final media with concentrations of 10 µM CNQX or 50 µM D-APV. In order to evaluate the effects of GLU or GLU-D and inhibitor for glutamate receptors, one culture was used in one experiment and was not used again.

### 2.3. Raman Spectroscopy

The Raman spectra of GLU/GLU-D solutions were measured by a home-made Slit-scanning confocal Raman microscopy with 532 nm continuous wave laser light for excitation [47]. The laser beam (Millennia-eV, Spectra Physics, Santa Clara, CA, USA) was introduced into the inverted microscope (ECLIPSE Ti2-E, Nikon, Tokyo, Japan) and was focused to the position at 20 µm above the substrate by using a 40×/1.25 numerical aperture, water immersion objective (CFI Apo Lambda S 40XC WI, Nikon, Tokyo, Japan). The laser power at the focus plane was 3 mW/µm^2^. The backscattered Raman signals were collected by the same objective lens, passed through a long-pass edge filter (LP03-532RU-25, Semrock, New York, NY, USA) and focused onto the entrance of spectrometer (MK-300, Bunko Keiki, Tokyo, Japan). The width of entrance slit was set at 40 µm. The Raman signals were then dispersed by a grating and detected by a cooled charge-coupled device camera (Pixis 400B, Teledyne Princeton Instruments, Trenton, NJ, USA). A total of 400 Raman spectra were measured simultaneously in one exposure time of 180 s and averaged with an image analysis software package, Fiji (National Institutes of Health, Bethesda, MD, USA).

### 2.4. Extracellular Potential Recording

All electrophysiological experiments were conducted at room temperature. The MED64 basic (alpha MED scientific) was used as a multisite extracellular potential recording system. The extracellular potentials were simultaneously measured with 64 electrodes at 20 kHz-sampling rates. The electrical signals were amplified by 20000-fold and band-pass filtered at 100–2000 Hz. The amplified signals were digitized at 16 bits resolution for storage. The spontaneous activity of neuronal network in each culture dish was measured for 10 min. In each electrode, single positive or negative peaks crossing thresholds were detected as single spikes. The thresholds were set at ±8 times of the standard deviation (S.D.) of baseline noise for a given electrode in each 500 ms. Firing rates were evaluated as the summation of spike numbers from 64 electrodes in each second. We measured spontaneous activities for 10 min at a normal recording medium to determine the initial firing rate in each culture, and then spontaneous activities before applications of GLU/GLU-D were measured and compared. The firing rates recorded in the conditions with 10, 20, and 30 µM of GLU/GLU-D were normalized by the rate obtained in the normal recording medium without GLU/GLU-D. The synchronized burst firings (SBFs) were analyzed based on the procedures described in previous papers [22,48]. In brief, the firing rate was calculated in each 100 ms time bin. The threshold for determining SBFs was defined as three times of the S.D. of firing rates for all time bins. Finally, time bins with firing rates that exceeded the threshold were defined as SBFs.

Statistical data are presented as means ± standard error of means (S.E.M) in 4 cultures unless otherwise indicated, and firing rates in single electrodes on MEA were presented as mean means ± S.D. is each culture.

## 3. Results and Discussion

### 3.1. Raman-Spectra of GLU and GLU-D Solutions

We characterized the differences of molecular structure of GLU and GLU-D by using Raman spectroscopy. As plotted in Figure 3, the Raman-spectra of 50 mM GLU and GLU-D in PBS aqueous solutions were measured and averaged. The Raman spectrum obtained from GLU-D solution exhibited several peaks in the cell-silent region at the wavenumber range of 2100–2300 cm^−1^ (described as a red square in Figure 3), which is considered to originate from CD_2_ stretching vibrational modes, while the CH_2_ stretching vibrational modes appeared at around 2980 cm^−1^ in the spectrum of GLU solution (described as a blue square in Figure 3). Raman spectra contain background signals from solvents and substrates in the samples. Raman scattering light due to water broadly appears in the whole spectrum. The peaks of oxygen and nitrogen appear around 1555 cm^−1^ and 2332 cm^−1^, respectively. In previous studies, Raman peaks of CD_2_ stretching vibrational modes have been reported to appear around 2100–2300 cm^−1^ [49,50,51,52]. Our experimental results together with previous studies indicate that the specific signals of C-D bonds of GLU-D can be detected by using Raman spectroscopy.

### 3.2. Effects of Extracellular GLU and GLU-D on the Spontaneous Activity of Cultured Hippocampal Neurons

We measured the spontaneous activity of cultured neurons before GLU/GLU-D applications in order to define the initial firing rate of neuronal networks. The unnormalized firing rates in total 64 electrodes were 1180 ± 340 and 700 ± 200 before GLU and GLU-D applications, respectively, suggesting that a lot of spikes were produced in the neuronal networks. In order to analyze the spatial distributions of spikes, we evaluated the averaged firing rates per single electrodes on MEA as shown in Table 1. Those values show large deviations indicating that spikes are spread channel to channel and detected in a lot of electrodes on MEA in the initial states. Then, we tested the effects of extracellular GLU/GLU-D on the spontaneous activity of neurons and the array-wide global activity in neuronal networks. The observed firing rates and the patterns were different for the different concentrations of GLU/GLU-D. The firing rate was confirmed to increase as extracellular concentrations of GLU/GLU-D increased (Figure 4a). The normalized firing rates recorded at different concentrations of GLU/GLU-D were shown in Figure 4b. The normalized firing rates were 1.41 ± 0.13, 1.85 ± 0.22, and 2.40 ± 0.27 for 10, 20, and 30 µM of GLU, respectively. For GLU-D, the values were 1.35 ± 0.14, 1.72 ± 0.18, and 2.18 ± 0.22 for 10, 20, and 30 µM of GLU-D, respectively. Thus, the spontaneous firing rates significantly increased with increasing extracellular concentrations of GLU and GLU-D. There was no significant difference between the dose-dependent changes of firing rates in the media with GLU and GLU-D. Previous studies have shown that extracellular GLU induces the influx of extracellular Ca^2+^ through the glutamate receptors [53,54,55]. The frequency of spontaneous electrical activity of cultured neurons was increased within a second [25]. Deuterium oxide that had been replaced with normal water did not affect the spontaneous firing rates in cortical neurons in vivo [56]. Our results confirmed that extracellular GLU-D enhances the spontaneous firing similarly to the effect of extracellular GLU on the spontaneous firing rates in cultured neuronal networks.

Firing rates significantly increased with increasing extracellular concentrations of GLU and GLU-D. In dissociated culture neurons, synchronized burst was observed after the development of neuronal circuits in the culture [17,18,19]. Bursting activity has been observed in vivo neuronal network, and that has been implicated for various phenomena such as synaptic plasticity and communications of neurons [57]. We evaluated the time series of array-wide firing rates in each 100 ms time windows, as shown in Figure 5a. Firing rate in each bin was confirmed to increase with increasing concentrations of GLU and GLU-D. In this experiment, spontaneous activity in each bin often decreased temporally during the recording period. In neuronal cells, electrical activity is known to be suppressed with inactivation of the voltage-dependent Na^+^ channels known as refractory periods displayed after production of action potentials. A lot of neurons are considered to be simultaneously suppressed by refractory periods, because the most neuronal activities in neuronal cells are enhanced by GLU/GLU-D applications. Therefore, the time-series of firing rates are thought to be fluctuated in recording period. The array-wide firing rates in SBFs each recording condition were normalized by the maximum values of each recording period. We selected the bin with the maximum firing rate during the recording time and normalized the value. In this data, spike densities in each 100 ms increased with increasing concentrations of GLU/GLU-D. To examine the effects of GLU/GLU-D on the neuronal network activity, firing rates in SBFs were analyzed as shown in Figure 5b. The normalized firing rates in SBFs were 1.16 ± 0.10, 1.60 ± 0.17, and 1.78 ± 0.25 for the conditions with 10, 20, and 30 µM of GLU, respectively. The values were 1.06 ± 0.11, 1.35 ± 0.05, and 1.96 ± 0.31 for the conditions with 10, 20, and 30 µM of GLU-D, respectively. The spike frequency in bursts increased with the increasing extracellular concentrations of GLU and GLU-D. Significant differences were not observed between the dependence of changes in firing frequency at different concentrations of GLU and that for GLU-D. An application of extracellular GLU evoked the excitatory post-synaptic potentials which lead to the generation of action potentials [58,59]. Hence, in the whole neuronal networks, excitatory post-synaptic inputs are activated by the application of extracellular GLU and the synchrony of neuronal firing is considered to be increased through enhanced synaptic transmissions [60]. A previous study using neuronal networks of primary cultured cortical neurons on MEA has shown the inhibitory effect of extracellular GLU for generating spikes when the applied concentration was more than 20 μM of GLU [26]. The inconsistency of the effects of extracellular GLU for cultured neurons and those for cortical neurons might be due to the difference of excitatory/inhibitory synapses ratio in the neuronal networks and that of the cortical networks in primary cultures. As future perspectives, the spatial burst pattern repertoires and stabilities should be derived to evaluate the similarity of spontaneous activities mediated by GLU and GLU-D. In hippocampal networks, nNOS connected with GABAergic neurons (nNOS inhibitory neurons) were more highly expressed compared to the nNOS connected with glutamatergic neurons (nNOS excitatory neurons), and that nNOS excitatory/inhibitory neurons ratio is different from the cortical networks [30]. Thus, nNOS connecting glutamate receptors are rarely activated in the hippocampal neurons due to low expressions of nNOS excitatory neurons, and firing rates in bursts increase with GLU/GLU-D concentrations.

### 3.3. Effects of Glutamate Receptors Blockers on Extracellular Glutamate- Induced Neuronal Activity

The above results showed that spontaneous firing of neurons and the synchronized firing of the neuronal networks changed depending on the concentrations of GLU-D equall to GLU. These results suggest that GLU-D is considered to be a useful analog of GLU with equivalent functions for the neuronal activity. We used 6-cyano-7-nitroquinoxaline-2,3-dione (CNQX), a potent and selective blocker of AMPA-type glutamate receptors and (*2R*)-amino-5-phosphonovaleric acid (D-APV), a selective blocker of NMDA-type glutamate receptors for pharmacological experiments. Intracellular [Ca^2+^] elevation initiated by the application of extracellular GLU was suppressed by CNQX and D-APV [61]. Thus, the effects of extracellular GLU-D might be inhibited by the treatment of these blockers if GLU-D activates neurons through glutamate receptors. We monitored the spontaneous activity of cultured neurons without extracellular GLU/GLU-D before the pharmacological treatments. Then GLU/GLU-D with CNQX or D-APV were added into the neurons. The obtained firing rates after the pharmacological treatments were normalized by that GLU/GLU-D value before the treatments, as shown in Figure 6. The firing rates drastically decreased in the presence of 10 µM CNQX. There were no significant changes of the firing rates in the conditions with different concentrations of GLU/GLU-D. The normalized firing rates were 0.24 ± 0.10, 0.20 ± 0.02, and 0.28 ± 0.05 for the conditions with 10, 20, and 30 µM of GLU, respectively, in the presence of 10 µM CNQX. The values were 0.28 ± 0.07, 0.24 ± 0.05, and 0.20 ± 0.03 for the conditions of 10, 20, and 30 µM of GLU-D, respectively, in the presence of 10 µM CNQX. On the other hand, the inhibitory effect of D-APV on the firing rates was moderate. We observed the increase of firing rates as the increase of applied concentration of GLU or GLU-D. The normalized firing rates were 0.97 ± 0.11, 1.18 ± 0.19, and 1.41 ± 0.13 for the conditions with 10, 20, and of 30 µM of GLU, respectively, in the presence of 50 µM D-APV. The values were 1.08 ± 0.06, 1.16 ± 0.12, and 1.31 ± 0.11 for the conditions of 10, 20, and 30 µM of GLU-D, respectively, in the presence of 50 µM D-APV. In spite of increasing concentrations of GLU or GLU-D, spontaneous firing was not increased in the presence of these blockers for glutamate receptors. The exposure time of extracellular solutions at 10, 20, and 30 μM of GLU and GLU-D was the same for spontaneous activity recordings with and without inhibitors of glutamate receptors (CNQX and D-APV) in the cultured hippocampal neurons. Under these conditions, spontaneous activity increased with increasing concentrations of GLU/GLU-D without inhibitors, so toxicities were not considered to be expressed in neuronal networks. Therefore, experimental data described in Figure 6 are considered to be the effects of only pharmacological treatments for glutamate receptors. Previous studies have reported that intracellular Ca^2+^ oscillations were suppressed with CNQX and D-APV treatments [62,63]. Especially, after CNQX treatments, inhibitory post synaptic currents evoked and the spontaneous activity was strongly inhibited in the neuronal network [64]. In the presence of CNQX, excitatory synaptic inputs were considered to be strongly suppressed and extracellular GLU/GLU-D cannot activate neuronal activity. On the other hand, in the presence of D-APV, firing rates were not suppressed, because the excitatory synaptic inputs mediated by non-NMDA receptors were maintained. The firing rates did not increase significantly due to the suppressed Ca^2+^ elevations mediated by NMDA receptors. From these results and previous studies, the suppression of dose-dependent GLU-D effects on neuronal activity by glutamate channel blockers support the idea that GLU can be replaced with GLU-D for monitoring the extra- and intra-cellular dynamics of GLU in the neuronal networks.

We monitored electrophysiological responses of primary cultured neurons exposed to GLU-D short-term, which were as short as 30 min. In living animals, the concentrations of GLU have been reported at around 0.6 μM in the synaptic regions at the resting conditions [65], and these concentrations have been shown to exceed 10 μM for 1–2 ms during the synaptic transmissions [31]. A total of 30 μM of GLU/GLU-D, which was a maximum concentration we used for this study, is close to the concentration at the synapse of neurons in physiological conditions in vivo. Because of this reason, spontaneous activity in the neuronal networks needs to be analyzed at the condition in the presence of GLU-D as high as 30 μM. On the other hand, long-term exposure to such high concentration of GLU leads to excitotoxicity in neurons [53,54,66]. In primary cultured neurons, approximately 20% of the neurons died after continuous exposure to 50 μM of GLU for a day [66,67]. For these reasons described above, we need to pay special care for long-term exposure to high concentration of GLU-D.

The combination of GLU-D treatment and Raman imaging can be a critical method to quantify the molecular concentration of GLU-D, and visualize their spatial distribution inside neuronal /glial cells. GLU-D can be recognized with minimal interference from the endogenous biomolecules by detecting CD_2_ stretching vibrational modes in silent region where no endogenous biomolecules have the Raman scattering signals. In summary, visualization of glutamate distributions by using GLU-D will provide powerful approaches for understanding the functions of excitatory synapses in the brain.

## 4. Conclusions

We studied the deuterium-labeled glutamate mediated neuronal activity of primary cultured rat hippocampal neurons and the networks. We confirmed that the district Raman peaks of CD_2_ stretching vibrational mode from GLU-D appeared at the wavenumber range of 2100–2300 cm^−1^ in the cell silent region. This data indicated that the intracellular glutamate concentration can be determined by measuring the signals of C-D bonds in cultured neurons, if GLU-D is uptaken inside of cells and works as same as GLU in a neuronal network. Next, we analyzed the effects of extracellular GLU-D on the spontaneous electrical activity and the global network activity of cultured neurons by using microelectrode array. The firing rates and the spike frequency in SBFs increased with increasing extracellular concentrations of GLU-D. These effects were suppressed by treatments with glutamate receptors antagonists. In the hippocampal neuronal networks, substantials differences in GLU/GLU-D mediated spontaneous activities were not confirmed. These results suggest that GLU-D is expected to work as an analog of GLU with equivalent roles in cultured neuronal networks, and that is the promising tracer to quantify the spatial distributions of GLU. Therefore the GLU-D has a potential application to visualize the dynamic properties of glutamate in the nervous systems, and the relationship between electrical responses and neurotransmitters dynamics inside/outside neuronal cells is expected to be revealed by combining the measurement methods of neuronal activity and visualization methods of deuterium localizations.

## Figures and Tables

**Figure 1 micromachines-11-00830-f001:**
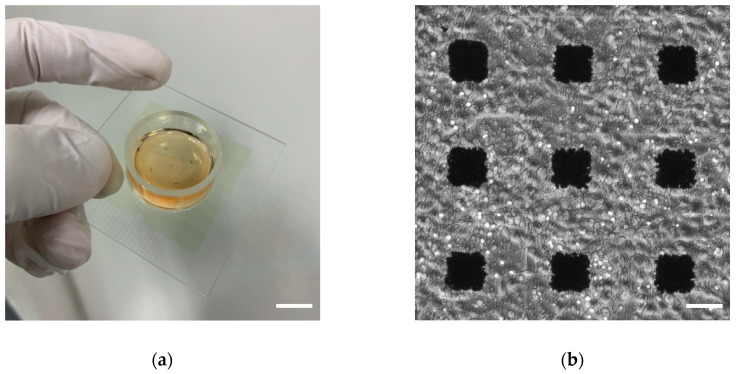
Culture of rat hippocampal neuronal network. (**a**) A photograph of the MED probe. Scale bar, 1 cm. (**b**) A phase-contrast micrograph of cultured rat hippocampal cells on the MED probe at 21 DIV. Black squares are electrodes. Scale bar, 50 μm.

**Figure 2 micromachines-11-00830-f002:**
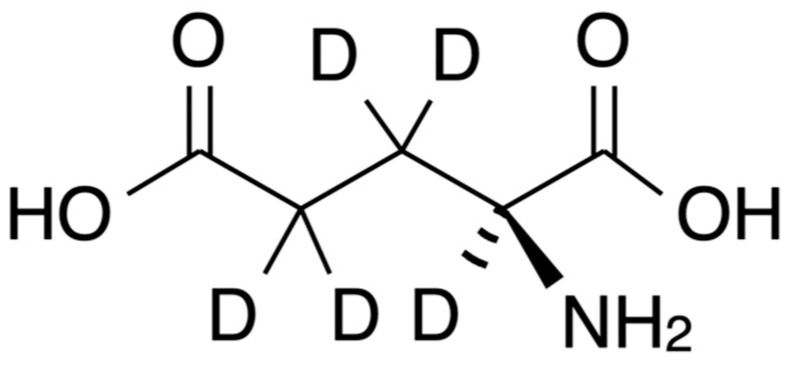
Chemical structure of L-glutamate-d5 (GLU-D).

**Figure 3 micromachines-11-00830-f003:**
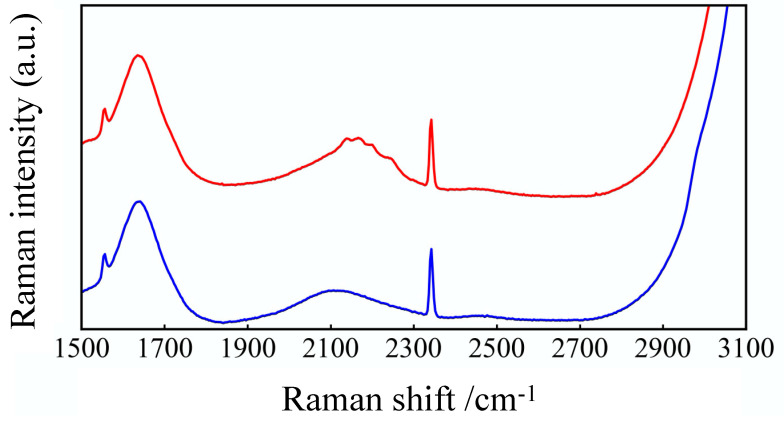
Raman spectra of GLU and GLU-D solutions. Blue and red lines show spectra of GLU and GLU-D, respectively. The peaks of oxygen and nitrogen appeared at around 1555 and 2332 cm^−1^, respectively. Red and blue squares correspond to wavenumber range of appearing CD_2_ stretching vibration of GLU-D and CH_2_ stretching vibration of GLU, respectively.

**Figure 4 micromachines-11-00830-f004:**
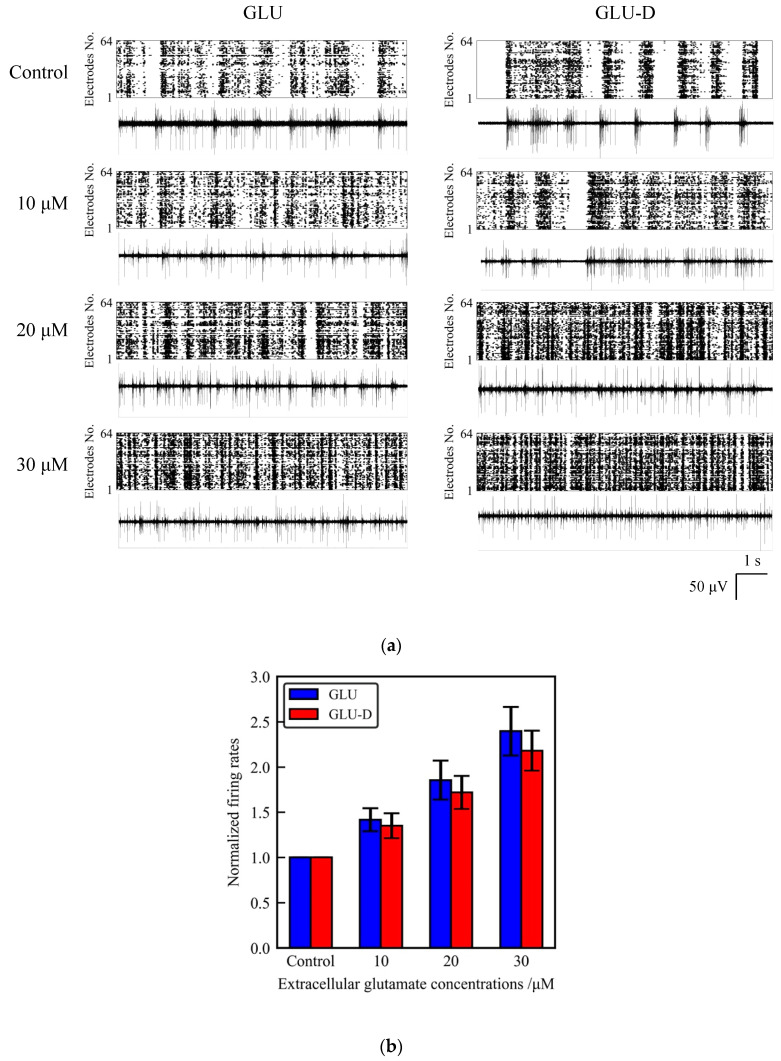
The effects of extracellular GLU/GLU-D on the spontaneous activity of primary cultured neurons at 30 DIV (for GLU) and 31 DIV (for GLU-D). (**a**) Spontaneous activity of neurons in the media with different concentrations of GLU/GLU-D. Upper panels are raster plots of 64 channels and lower panels are the typical waveforms. (**b**) Normalized mean firing rates at each GLU (blue) and GLU-D (red) concentration.

**Figure 5 micromachines-11-00830-f005:**
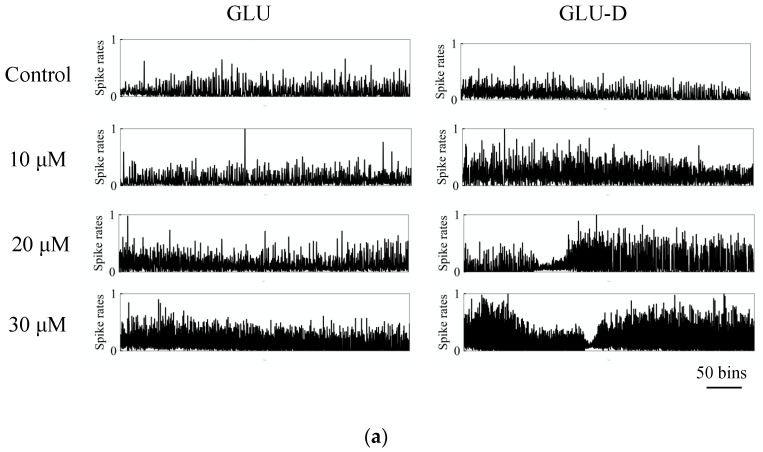
The effects of extracellular concentrations of GLU/GLU-D on the synchronous firing activity of the neuronal networks at 30 DIV. (**a**) Typical waveform of normalized spike frequency in each 100 ms time bin. (**b**) Normalized mean spike frequency in SBFs at each concentration of GLU (blue) and GLU-D (red).

**Figure 6 micromachines-11-00830-f006:**
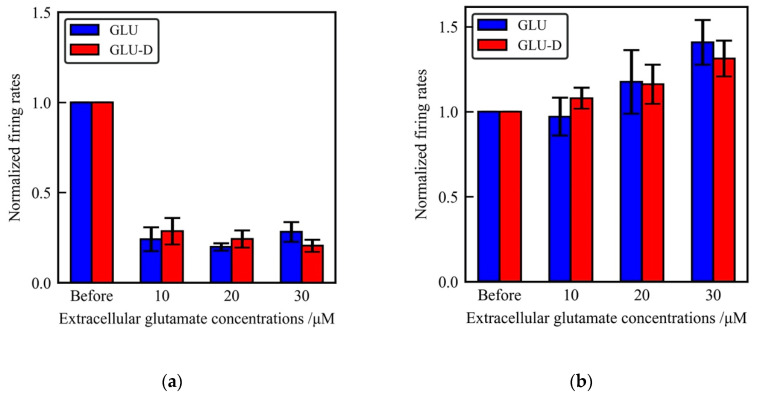
The effects of glutamate receptors blockers on extracellular glutamate-induced neuronal activity. (**a**) Mean firing rates at different concentrations of GLU (blue) and GLU-D (red) in the media with 10 µM of CNQX. (**b**) Mean firing rates at different concentrations of GLU (blue) and GLU-D (red) under 50 µM of D-APV.

**Table 1 micromachines-11-00830-t001:** Averaged firing rates at single electrode on MEAs before GLU/GLU-D applications. These cultures (22–37 DIV) were used to measure spontaneous activity. The data were shown as Mean ± S.D.

Culture No.	Before GLU Application (spikes/s)	Before GLU-D Application (spikes/s)
Culture No.1	36.7 ± 24.2	16.9 ± 12.8
Culture No.2	14.6 ± 9.0	2.3 ± 3.0
Culture No.3	11.7 ± 12.2	8.0 ± 5.8
Culture No.4	10.2 ± 6.7	16.3 ± 12.3

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
