# Peer review of "Deuterated Glutamate-Mediated Neuronal Activity on Micro-Electrode Arrays"

_micromachines, 2020, doi:10.3390/mi11090830_

Round 1

Reviewer 1 Report

This manuscript presents a study of the effects of extracellular deuterium-labeled glutamate (GLU-D) on neuronal activity measured in rat hippocampal neurons. The sum of the results suggests that GLU-D can be employed as a glutamate analog in monintoring neural dynamics, however it does remain to be seen how consistent these results can be in various forms of cultured networks.

The authors present a compelling story using Raman peaks of the CD2 stretching vibrational mode of GLU-D to demonstrate that glutamate concentration is linked to C-D bond signals. Furthermore, the authors show that GLU-D and GLU significantly increase spontaneous firing rates.

When assessing synchronous firing activity, the authors found that spike frequency in burst increased with increasing extracellular GLU and GLU-D, however there was no observation of a significant difference. Furthermore, the authors did not observe what has been reported regarding GLU inhibitory dynamics at for cortical neurons at higher extracellular concentrations--attributing it to the difference in the ratio of excitatory/inhibitory synapses. Many reports have indicated that the aforementioned effect of GLU in some cortical cultures requires the presence of nNOS inhibitory neurons, and the enhanced susceptibility in these cultures is likely due to adequate balance of excitatory and inhibitory neurons (nNOS neurons included). Thus, the differences in ratios observed here, while likely a result of differing ratios, can possibly be controlled with more intentional inspection of the neuronal networks.

The authors went on to monitor the spontaneous activity of cultured neurons in the presence of GLU channel blockers, finding suppressed dose-dependent GLU-D neuronal activity effects by GLU channel blockers. One striking observation was that neurons died (approx 20%) after continuous high GLU/GLU-D exposure (over 50 uM), which may obscure some differential effects of GLU versus GLU-D. While GLU-D doesn't appear to be substantially different than GLU in promoting firing rates and spike frequency in the presence and absence of glutamate receptor antagonists, the evidence is not particularly compelling that GLU-D can mechanistically serve as a viable analog for GLU, particularly in the context of inconsistencies in the observed data relative to reported trends and toxicity of prolonged exposure.

The data showing the utility of GLU-D in minimizing endogenous biomolecule interference is compelling, however the data showing no substantial differences in neural activity leaves a little to be desired.

It is recommended that, if possible, the authors further address the concerns of feasibility and incongruent experimental results more thoroughly, particularly in the conclusions section.

Author Response

Dear the Reviewer 1,

Our point by point responses were wrote in word file.

Please see the attachment of our responses.

Best regards,

Wataru Minoshima, Kyoko Masui, Tomomi Tani, Satoshi Fujita, Hidekazu Ishitobi, Chie Hosokawa, and Yasushi Inouye.

Reviewer 2 Report

The authors present the use of deuterated Glu (GLU-D) to track neurotransmitter signalling in neurons in vitro. To ensure that the tracking material does not alter activity of the cell in ways different than non-deuterated Glu, activity rates in different concentrations of Glu and GLU-D are presented. Activity changes are also tested in the presence of receptor blockers to confirm excitatory action via the receptor. The Raman spectrum of GLU-D is shown distinguishable from the Glu spectrum. However, to support the authors' main claim that GLU-D can be used as an intracellular tracer, the distinguishability of GLU-D from the cell remains to be shown. The argument that GLU-D has similar physiologic effects as Glu is supported by the experiments provided.

Comments

Line 48-50, MEA experiments to understand neuronal signaling is a very active field. A statement of their use within the last 10 years, or reference to one of the many reviews available in the last 10 years would be appropriate.

Line 110-112, It is unclear how the medium is changed. Please be more explicit. Is the first plating with half glia conditioned medium and the subsequent feedings remove half of the existing mixture and add half of one chip volume of new medium, or is the medium that is removed replaced with 50% new medium (now .25 of the chip volume) + 50% conditioned medium (now .25 of the chip volume)?

Line 125-131, The data suggests that baseline measurements were made before addition of GLU or GLU-D to the measurement medium, but this is not described in the methods.

What is the relationship between chip and measurement medium? Is one chip measured only in control and one concentration, or are multiple concentrations applied serially? Can GLU effects be washed out fast enough to test activity change of GLU vs. GLU-D on the same culture? For example baseline recording with no additives, record with GLU, record with no additives until baseline is re-established, record with GLU-D?

Figure 4a, the single channel traces are too zoomed out to show the waveforms recorded and determine what relevant properties you aim to highlight. Raster plots for 20µM and 30µM conditions are also too much data to be meaningfully displayed in the document. Is a smaller time window still exemplary of what you aim to show? Is a scrollable electronic supplement an option to include the large data set in a meaningfully viewable way? What is the take-home message of showing the example recordings? Do you present the spike form to compare the progression of a GLU vs. a GLU-D action potential? Do you present the bursting data to compare changes in spike form during GLU vs. GLU-D bursts? Is a comparison of the fraction of spikes in a burst vs. outside a burst relevant? For example, 9 spikes may be spread within 1 second of recording as individual spikes at 9 Hz, or as 3 small bursts with 3 spikes each at 100 Hz interspersed with 470ms in between. Each of these may or may not be the goal, the data is in too much of a dark block to see.

Within one culture, what is the spread of firing rates from channel to channel? This may be already contained in the raster plots, but at the size in the PDF it cannot be seen.

Line 214-216, Bursting activity has been documented in the literature for in vitro neural cultures as well, and those instances are more similar to the system presented here.

Line 217, Why were activity rates not normalized to the conditions before GLU or GLU-D were added?

Line 221, if firing rates are normalized to (divided by) the maximum rate, how are values greater than 1 achieved?

The data trends are the same for GLU and GLU-D, but except for one  instance the amplitude of the GLU-D effect is consistently lower. Mean +/- SD or scatter plots would clarify if this is a trend or just noise. SEM only indicates how close the mean is to the population mean.

It is somewhat surprising that the investigation does not investigate more of the parameters described in the work of Han et al, despite citing it.

Figure 5 caption, What is binGLUD3? If this is the name of the specific bin from which the traces are shown, the trace should be 100ms long. As noted previously, with this zoom and print size, waveforms cannot be determined from the traces as shown. If 5a shows the firing rate of each bin then the horizontal scale should be in number of bins instead of seconds, and please comment on the fluctuations in rate within one sample during constant GLU or GLU-D concentration, for example the steep periods of reduced activity in 20 and 30µm GLU-D. If they must be normalized, the spike rates should be normalized to the average firing rate, or average bursting firing rate, in the control condition prior to manipulation. Normalized rates greater than 1 would then indicate increased activity and less than 1 decreased activity.

Line 273, to make your point from the abstract: 'GLU-D can be replaced with GLU for monitoring...' should be 'GLU can be replaced with GLU-D for monitoring...'; The GLU-D is in the monitored sample.

Line 300-302 The final conclusion is overstated. Proof of the detection has not been shown here. Can the GLU-D be detected in the background of the cellular components instead of just in buffer? The Raman spectrum of a neuron vs. the GLU and GLU-D in Fig. 3 would be useful in lieu of a cell measurement of GLU-D.

Minor

Line 47, planer --> planar

Line 102, On the MED probe, 64, 50 x 50 µm planar microelectrodes

Line 105 and throughout, (noun) was consisted of (noun) --> (noun) consisted of (noun)

Line 177, considered to originate from

182 reported to appear around

Paragraph starting line 212 please check grammar.

Author Response

Dear the Reviewer 2,

Our point by point responses were wrote in word file.

Please see the attachment of our responses.

Best regards,

Wataru Minoshima, Kyoko Masui, Tomomi Tani, Satoshi Fujita, Hidekazu Ishitobi, Chie Hosokawa, and Yasushi Inouye.
